# Gender-Related Differences in the Correlation between Odor Threshold, Discrimination, Identification, and Cognitive Reserve Index in Healthy Subjects

**DOI:** 10.3390/biology12040586

**Published:** 2023-04-12

**Authors:** Carla Masala, Paolo Solla, Francesco Loy

**Affiliations:** 1Department of Biomedical Sciences, University of Cagliari, SP8 Cittadella Universitaria Monserrato, 09042 Cagliari, Italy; 2Department of Neurology, University of Sassari, Viale S. Pietro 10, 07100 Sassari, Italy

**Keywords:** olfactory function, smell, Sniffin’ Sticks, olfaction, cognitive reserve index

## Abstract

**Simple Summary:**

Many studies indicated significant associations between olfactory function and cognitive abilities in healthy controls. However, the gender-related association between olfactory function and each specific cognitive domain of the Cognitive Reserve Index (CRI) questionnaire has so far not been evaluated. The aim of this study was to evaluate gender-related differences in the relationship between olfactory function and each specific cognitive domain of the CRI questionnaire, such as education, working activity, and leisure time in healthy subjects. Our data indicated significant gender-related associations between olfactory function and CRI score. In women, odor threshold, odor discrimination, and identification were associated with CRI-Leisure Time, while in men, only a significant association between odor threshold and CRI-Education was observed.

**Abstract:**

Background: Many studies suggested that olfactory function could be associated with semantic memory, executive function, and verbal fluency. However, the gender-related association between olfactory function and the cognitive domain is not well investigated. The aim of this study was to estimate gender-related differences in the relationship between olfactory function and each specific cognitive domain of the Cognitive Reserve Index (CRI) questionnaire, such as education, working activity, and leisure time in healthy subjects. Methods: Two hundred and sixty-nine participants were recruited (158 women and 111 men), with a mean age of 48.1 ± 18.6 years. The CRI questionnaire and Sniffin’ Sticks test were used to evaluate the cognitive reserve and the olfactory function, respectively. Results: In all subjects, significant associations between the odor threshold versus CRI-Education, between the odor discrimina-tion and identification versus CRI-Working activity and CRI-Leisure Time, were found. In women, odor threshold, discrimination, and identification were associated with CRI-Leisure Time, while in men, only a significant association between odor threshold and CRI-Education was observed. Conclusions: Our data, showing significant gender-related associations between olfactory function and CRI scores, suggested the use of olfactory evaluation and cognitive reserve as an important screening tool for the early detection of mild cognitive impairment.

## 1. Introduction

In the literature, recent advances in cognitive abilities studies in relation to well-being, social interactions, and sport activities have been observed; in particular, some research focused on the role of chemosensory perception [1,2]. In fact, olfactory function could act on cognitive abilities as related to semantic memory, executive function, and verbal fluency [3,4,5]. Moreover, a low language ability has been associated with low olfactory function [3,4,5]. Olfactory function plays an important role in the regulation of emotional function, social interactions, and eating behavior. The decreased olfactory function may show negative implications for human well-being and social relationships.

Olfactory function and cognitive performance showed an overlap in neural substrates and were associated with the orbitofrontal cortex, hippocampus, and entorhinal cortex [3,4,5]. This relationship suggested the need to evaluate olfactory function as a potential early biomarker in patients’ cognitive impairment [6,7,8,9]. Recently, the importance of olfactory assessment has significantly increased, particularly in cases of mild cognitive decline and in the diagnosis of COVID-19.

The objective evaluation of quantitative olfactory dysfunction comprises three different components: odor threshold (OT), odor discrimination (OD), and odor identification (OI) [10]. The OT is usually more associated with individual differences in nasal anatomy [11], and it is related to a specific cognitive domain, such as language function [6]. Indeed, OD and OI are considered in relation to the subcortical part of the olfactory system and are related to the ability to differentiate and identify different odors, respectively [12]. Usually, lower scores in cognitive tests associated with a deficit in olfactory function may suggest a mild cognitive impairment [13]. In this context, the brain’s cognitive reserve is closely related to cortical plasticity and is considered the potential capability of the brain to cope with neuronal damage in relation to individual differences such as brain size and synapse count. Cognitive reserve plays an important role in order to recover brain damage affected by aging or neurodegenerative diseases through the recruitment of pre-existing brain networks. A high score of CRI indicates better neuronal plasticity to compensate for brain atrophy and is considered a protective factor, while a lower score indicates brain vulnerability. In general, setting aside neurocognitive resources during the lifetime may preserve the brain from mild cognitive impairment and pathologies such as Alzheimer’s disease [14,15]. Since around 65% of subjects affected by Alzheimer’s disease are women [16], understanding gender-related differences in the relationship between olfactory function and the specific cognitive domain of the CRI questionnaire appears important in order to obtain the most incisive interventions for an early diagnosis of mild cognitive impairment and Alzheimer’s disease. Since the olfactory deficit is considered an important health issue, which may play a significant role in human social interactions and well-being, in our study, we hypothesized the occurrence of correlations between olfactory function and CRI in healthy subjects in relation to sex. In the literature, few studies have evaluated the gender differences in CRI [17,18], but the gender-related association between olfactory function and each specific cognitive domain of the CRI questionnaire has so far not been evaluated.

The aim of this study was first to evaluate gender-related differences in each specific cognitive domain of the CRI questionnaire such as education, working activity, and leisure time in healthy subjects. Then, we evaluated gender-related differences in the relationship between olfactory function and each specific cognitive domain of the CRI.

## 2. Materials and Methods

### 2.1. Participants

Two hundred and sixty-nine participants were recruited (158 women with a mean age of 46.3 ± 18.6 years and 111 men with a mean age of 50.4 ± 18.4 years) with an age range of 20–85 years and a mean age of 48.1 ± 18.6 years. Data were collected from May 2018 to January 2020. Exclusion criteria were neurodegenerative diseases, thyroid disorders, chronic renal diseases, a history of head or neck trauma, nasal pathology, acute respiratory infections, diabetes, stroke, and any systemic disease associated with smell disorders. The clinical evaluation for each participant included age, sex, body mass index (BMI), current medications, smoking history, and employment. For each subject, olfactory and cognitive function were assessed.

### 2.2. Ethical Standard

This study was approved by the local Ethics Committee (Prot. PG/2018/10157) on 28 March 2018 and was performed according to the Declaration of Helsinki. All participants received an explanatory statement and gave their written informed consent to participate in the study.

### 2.3. Procedures

In all participants, olfactory function was evaluated using the Sniffin’ Sticks test (Burghart Messtechnik, Wedel, Germany), which assessed three different parameters: OT, OD, and OI [10,19,20]. According to the Sniffin’ Sticks guideline, first, OT was determined using 16 stepwise dilutions of n-butanol with a single-staircase technique based on a three-alternative forced-choice (3AFC) task. Second, OD was measured over 16 trials [21,22,23]. In the discrimination test, three pens were presented, two containing the same odor and the third containing the target odorant (3AFC task). Third, OI was measured using 16 common odors, each presented with four verbal descriptors in a multiple forced-choice format (three distractors and one target). The interval between each odor presentation was 20–30 s. Then, we calculated the total score of olfactory function (Threshold, Discrimination, Identification, TDI), and functional anosmia, hyposmia, normosmia, and supersmellers were indicated by a score ≤16, between 16.25 and 30.5, between 30.75 and 41.25, >41.5, respectively [20]. In addition, the cognitive reserve was quantified by using the CRI score [24]. The self-reported questionnaire measures the amount of cognitive reserve acquired during a participant’s lifetime. The CRI questionnaire consists of 20 items grouped into three main sources: education, working activity, and leisure time. Each of these items in a subject’s lifetime is calculated as a sub-score. The CRI-Education evaluated years of education and the possible training courses lasting six months and the total score was the sum of these factors. The CRI-Working Activity evaluated the number of years in each profession during the lifespan, and the total score was the sum of the working activity years multiplied by the cognitive level of the job from 1 to 5 years. The CRI-Leisure Time evaluates the frequency and number of years spent in intellectual activities. The total score was computed as the total number of years involved in these activities in which frequency is often/always.

Moreover, the cognitive abilities of each participant were also assessed with the Montreal Cognitive Assessment (MoCA), which evaluates different domains: visual-constructional skills, executive functions, attention and concentration, memory, language, conceptual thinking, calculations, and spatial orientation [25,26].

### 2.4. Statistical Analysis

The analysis was carried out using SPSS 26.0 for Windows (IBM, Armonk, NY, USA). All data were presented as mean values ± standard deviation (SD). Statistical differences between men and women were calculated using one-way ANOVA. In order to discover the more promising factors for the multivariate linear regression analyses, bivariate correlations between each specific cognitive domain of the CRI questionnaire (Education, Working Activity, and Leisure Time) and the olfactory function using Pearson’s correlation coefficient were performed. The multivariate linear regression analysis was performed in different models using OT (model 1), OD (model 2), and OI (model 3) as dependent variables, while CRI-Education, CRI-Working Activity, and CRI-Leisure Time were independent variables. A *p*-value < 0.05 was considered statistically significant.

## 3. Results

The demographic and clinical information of all participants was indicated in Table 1. Significant differences between men and women were observed for height, weight, OT, OI, TDI score, cognitive abilities (MoCA scores), and CRI-Working Activity. In particular, women exhibited higher mean values in OT [F_(1,267)_ = 6.031, *p* < 0.05, partial η^2^ = 0.022] and TDI score [F_(1,267)_ = 5.774, *p* < 0.05, partial η^2^ = 0.021], and lower significant scores in CRI-Working Activity (Table 1). No significant differences (*p* > 0.05) between both sexes for OD, OI, MoCA global score, CRI-Education, CRI-Leisure Time, and CRI-Total Score were found.

Considering all subjects, significant bivariate correlations were observed between OT and CRI-Education (r = −0.150, *p* = 0.014), while OD was significantly correlated to CRI-Working Activity (r = 0.401, *p* = 0.001), CRI-Leisure Time (r = 0.364, *p* = 0.001), and CRI-Total Score (r = 0.197, *p* = 0.001) (Table 2). Moreover, significant bivariate correlations were detected between OI and different CRI sub-domains such as CRI-Education (r = 0.176, *p* = 0.004), CRI-Working Activity (r = 0.406, *p* = 0.001), CRI-Leisure Time (r = 0.389, *p* = 0.001), and CRI-Total Score (r = 0.337, *p* = 0.001) (Table 2). The TDI score was significantly correlated to CRI-Working Activity (r = 0.308, *p* = 0.001), CRI-Leisure Time (r = 0.304, *p* = 0.001), and CRI-Total Score (r = 0.297, *p* = 0.001).

In addition, in order to better clarify the role of bivariate correlations, multivariate linear regression analyses were performed to predict olfactory dysfunction in relation to CRI-Education, CRI-Working Activity, and CRI-Leisure Time. The multivariate linear regression analyses showed that CRI-Education was a significant predictor when using OT as a dependent variable [F_(4,264)_ = 10.724, *p* = 0.002] (Table 3). The model with OT as a dependent variable explained 4% of the variance (R^2^ = 0.041) (Figure 1A). In the second model, CRI-Working Activity and CRI-Leisure Time were significant predictors for the OD [F_(4,264)_ = 16.790, *p* = 0.0001] and this model explained around the 20% of the variance (R^2^ = 0.203) (Figure 1B,C). Similarly, in the third model, significant associations were observed between CRI-Working Activity and CRI-Leisure Time and the OI [F_(4,264)_ = 16.986, *p* = 0.0001] and this model explained around 19% of the variance (R^2^ = 0.193) (Figure 1D,E).

In women, significant correlations were found between CRI-Working Activity and OD (r = 0.313, *p* < 0.001), and OI (r = 0.426, *p* < 0.001), and TDI score (r = 0.355, *p* < 0.001). Moreover, we also observed significant correlations between the CRI-Leisure Time and OT (r = 0.278, *p* < 0.001), and OD (r = 0.398, *p* < 0.001), and OI (r = 0.546, *p* < 0.001), and TDI score (r = 0.526, *p* < 0.001) (Table 4). The CRI-Education was significantly correlated only with the OI (r = 0.184, *p* = 0.020) and TDI score (r = 0.333, *p* < 0.001). Finally, the CRI-Total score was correlated with OD (r = 0.325, *p* < 0.001), OI (r = 0.337, *p* < 0.001), and TDI score (r = 0.298, *p* < 0.001) (Table 4).

In women, the multivariate linear regression analyses showed that in model 1, obtained using OT as a dependent variable, CRI-Leisure Time was significantly associated with OT [F_(4,153)_ = 3.514, *p* = 0.002] (Table 5), and the model explained 8% of the variance (R^2^ = 0.084 (Figure 2A). Similarly, in model 2, using OD as a dependent variable, a significant association was found between OD and CRI-Leisure Time [F_(4,153)_ = 8.396, *p* = 0.004] with the 18% of the variance (R^2^ = 0.180) (Figure 2B). While, in model 3, performed using OI as a dependent variable, two different significant associations were observed between CRI-Working Activity [F_(4,153)_ = 18.450, *p* = 0.001] and CRI-Leisure Time [F_(4,153)_ = 18.450, *p* = 0.018] and OI (Figure 2C,D), with around 33% variance (R^2^ = 0.325) (Table 5).

In men, the following significant correlations were found: between OT and CRI-Education (r = −0.346, *p* = 0.001), between OI and CRI-Education (r = 0.194, *p* = 0.042), between OD and CRI-Working Activity (r = 0.379, *p* = 0.001), between OI and CRI-Working Activity (r = 0.342, *p* = 0.001), between TDI score and CRI-Working Activity (r = 0.290, *p* = 0.002), between OD and CRI-Leisure Time (r = 0.333, *p* = 0.001), between OI and CRI-Leisure Time (r = 0.334, *p* = 0.001), between TDI score and CRI-Leisure Time (r = 0.205, *p* = 0.031), and between OI and CRI-Total Score (r = 0.387, *p* = 0.001) (Table 6).

Moreover, in men, multivariate linear regression analyses showed a significant association only between OT and CRI-Education [F_(4,106)_ = 4.440, *p* = 0.002] with the model explaining around 14% of the variance (R^2^ = 0.143) (Figure 3). Instead, no significant associations (*p* > 0.05) were found between OD and OI and each specific sub-score of CRI (Table 7).

## 4. Discussion

This study focused on the evaluation of gender-related differences in the association between olfactory function and cognitive reserve index. The brain cognitive reserve is closely related to cortical plasticity and is considered the potential capability of the brain to cope with neuronal damage in relation to individual differences such as brain size and synapse count. Cognitive reserve is important in order to recover brain damage affected by aging or neurodegenerative diseases through the recruitment of pre-existing brain networks [27]. Robertson indicated that the right hemisphere plays an important role in cognitive reserve using a noradrenergic pathway [28].

Our results showed statistical differences between men and women for the OT, global olfactory function (TDI score), and CRI-Working Activity. According to previous studies, in women, higher mean values in OT and TDI scores were found compared to men [29,30]. Moreover, in our study, women showed significantly decreased scores in CRI-Working Activity compared to men, while no significant differences were observed for CRI-Education and CRI-Leisure Time. This result may be explained as a difference in the employment and retirement age between the two sexes, as reported by Boots and Colleagues [31]. In fact, some studies reported that women with healthy working conditions (e.g., crafts worker, shopkeeper, and farmer) may reduce the risk of mild cognitive impairment and dementia [18,32].

Considering all subjects, our data showed a noteworthy association between olfactory function and CRI. In particular, we found significant associations between the OT and CRI-Education and between the OD and OI and CRI-Working Activity and CRI-Leisure Time. These data suggested and highlighted the close association between olfactory function and cognitive abilities. Subjects with lower scores in olfactory function usually exhibit weaker cognitive performance [2]. In addition, higher olfactory scores are usually associated with better semantic memory and verbal abilities [33]. Craick and colleagues [34] showed that, both in women and men, Alzheimer’s disease symptoms appeared five years later in bilinguals than in monolinguals. Another study indicated that cognitive ability and vocabulary were associated with OI [35]. A possible explanation of these data is due to partial overlapping in the brain areas involved in cognitive abilities and those involved in olfactory function such as the orbitofrontal cortex and amygdala. Our data also suggested a significant positive correlation between OD and OI scores and CRI-Leisure Time and are similar to those obtained in a previous study [36], suggesting a relationship between OI and social life. On the other hand, the relationship between OI and OD and CRI-Leisure Time is not clearly understood. In the multivariate linear regression analyses, we found that there was a positive significant association between OI and OD and CRI-Leisure Time only in women, but not in men. These data support the hypothesis that CRI-Leisure Time is closely correlated to OI and OD performance only in women. Generally, women showed better olfactory performance compared to men [10,37] and also had more social connections. It is likely that women perform differently in social relationships and there could possibly be an association between leisure time, social networks, and health measures. In fact, Codina and Pestana showed that men had more leisure time, but women had a higher positive leisure experience than men [38]; women enjoyed themselves more with less leisure time and were more positive about time orientations. Moreover, Larsson and colleagues observed that women identified more odors than men due to gonadal hormones, fluctuations of the menstrual cycle, and neuroendocrine influences on brain regions involved in olfactory function, but sex differences disappeared in older age [35]. Although the potential cause of the difference between men and women remains unclear, the higher identification in women may be due to sex differences in verbal abilities, prior experience, and odor memory [39]. However, social factors may also contribute as women generally experience greater olfactory pleasantness, odor familiarity, and greater exposure to odors in their social environment. A better performance in OD and OI is considered a measure of general good health in the population. Indeed, good health is often connected with social lives and the number of social contacts that the individuals have in their life.

The association between OT and CRI-Education was observed only in men and not in women. Our data suggested that the CRI-Education sub-score may have a minor contribution in women, as indicated in a previous study [17]. Instead, Heian and colleagues showed that men with low education had lower olfactory function scores after a comparison between self-reported tests and Sniffin’ sticks data analysis [40].

Our data suggested that in men, CRI-Education and not CRI-Working activity was associated with the odor threshold. Moreover, both in men and in women, working activity had no relation with the total olfactory function. Education probably has a protective role in mild cognitive decline, as previously indicated by Meng and D’Arcy [41]. In fact, people with a high level of education correctly identified more odors, as indicated by Larsson and colleagues [35].

Considering these results, our data suggested a potential role of biomarkers for olfactory function in the early diagnosis of mild cognitive impairment. Similarly, other previous studies indicated that olfactory impairment represents a peculiar biomarker in neurodegenerative disorders [6,7,8,42,43,44,45,46]. Recently, there has been an increased interest in the evaluation of olfactory dysfunction in the early stage of neurodegenerative disorders such as Parkinson’s disease. In our previous study on Parkinson’s disease, patients’ significant correlations were observed between OT and language, between OD and visuospatial domain, and between OI and executive index scores and attention [6], suggesting that the OT, OD, and OI are differently related to the cognitive abilities of the subjects.

One limitation of our study is the cross-sectional design, thereby it did not allow us to evaluate these associations over time.

## 5. Conclusions

Our data indicated gender-related associations between olfactory function and Cognitive Reserve Index. In women, odor threshold, odor discrimination, and identification were associated with CRI-Leisure Time, while in men, only a significant association between odor threshold and CRI-Education was observed. The gender differences observed in our study could play a key role in order to predict the risk of mild cognitive impairment and to develop a precision medicine approach. In fact, this study could help in the development of new and appropriate intervention strategies differentiated by sex regarding the prevention of cognitive impairment. Our study confirmed that olfactory dysfunction and cognitive impairment had a severe negative impact on subjects’ daily life. Finally, this study highlighted the use of olfactory evaluation and cognitive reserve assessment as important screening tools for the early detection of mild cognitive impairment.

## Figures and Tables

**Figure 1 biology-12-00586-f001:**
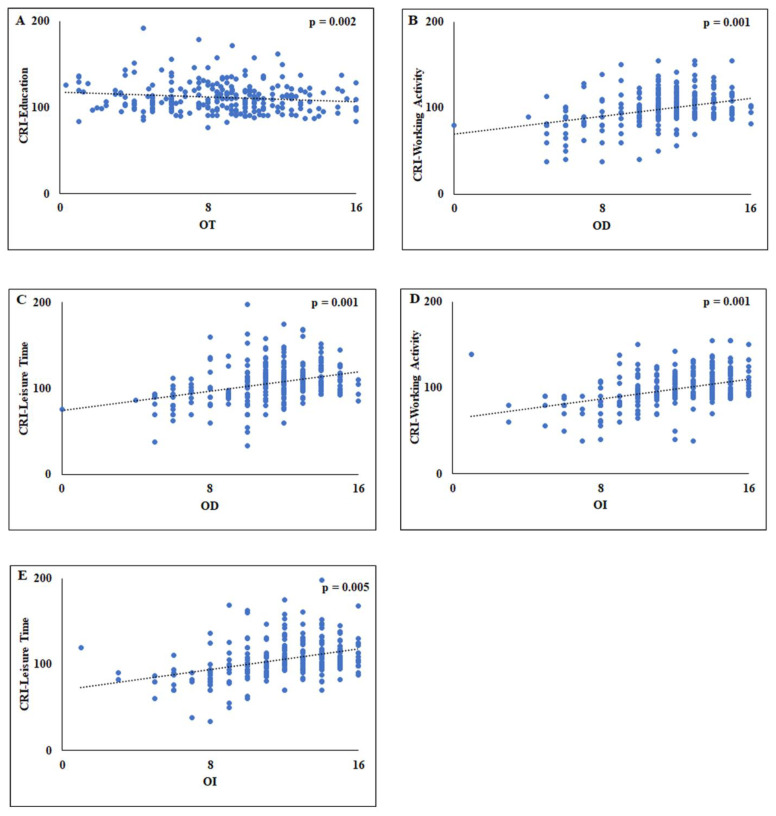
Scatterplots of the relationship between the OT and CRI-Education (**A**), between OD and the CRI-Working Activity (**B**), between OD and the CRI-Leisure Time (**C**), between OI and the CRI-Working Activity (**D**), and between OI and CRI-Leisure Time (**E**). CRI = Cognitive Reserve Index; OT = Odor Threshold; OD = Odor Discrimination; OI = Odor Identification.

**Figure 2 biology-12-00586-f002:**
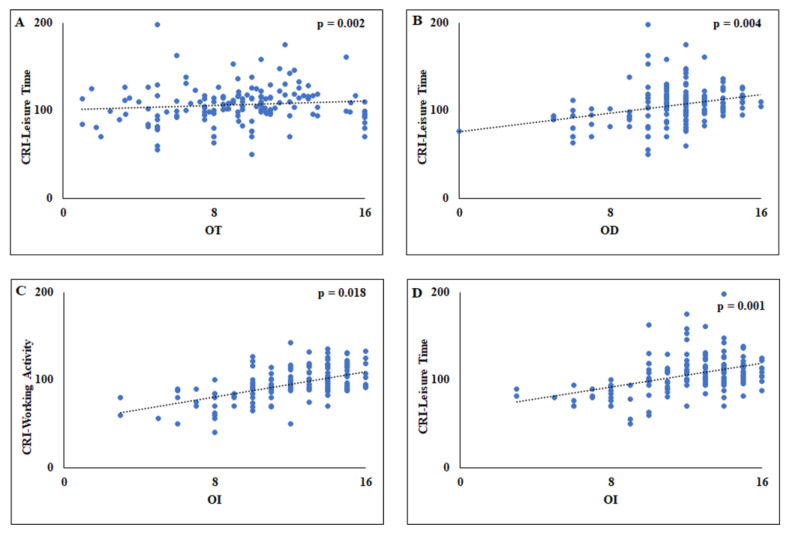
Scatterplots of the relationship between the OT and CRI-Leisure Time (**A**), between OD and the CRI-Leisure Time (**B**), between OI and the CRI-Working Activity (**C**), and between OI and CRI-Leisure Time (**D**). CRI = Cognitive Reserve Index. OT = Odor Threshold; OD = Odor Discrimination; OI = Odor Identification.

**Figure 3 biology-12-00586-f003:**
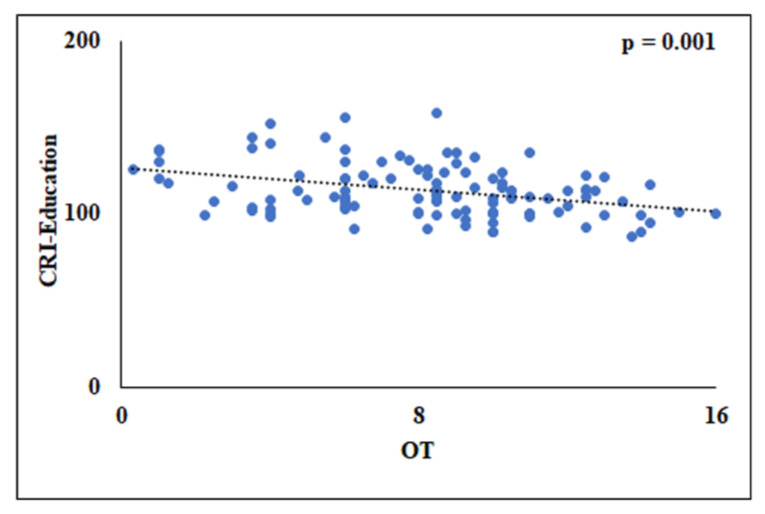
Scatterplot of the relationship between the odor threshold (OT) and CRI-Education in men (*n* = 111). CRI = Cognitive Reserve Index.

**Table 1 biology-12-00586-t001:** Demographic and clinical information of all participants. Data are indicated as mean score ± SD.

	Total	Men	Women	Significance
Age (years)	48.1 ± 18.6	50.4 ± 18.4	46.3 ± 18.6	*p* > 0.05
Height (cm)	1.67 ± 8.7	1.72 ± 8.5	1.63 ± 6.7	***p* < 0.001**
Weight (kg)	69.1 ± 14.5	83.9 ± 76.6	63.7 ± 11.5	***p* < 0.001**
OT	8.9 ± 3.6	8.3 ± 3.5	9.4 ± 3.5	***p* < 0.05**
OD	11.3 ± 3.1	10.9 ± 2.9	11.4 ± 2.6	*p* > 0.05
OI	11.9 ± 2.9	11.5 ± 2.9	12.3 ± 2.8	*p* < 0.05
TDI Score	32.1 ± 6.9	30.9 ± 7.1	33.1 ± 6.7	***p* < 0.05**
MoCA	26.5 ± 2.9	26.9 ± 2.9	26.7 ± 2.9	*p* > 0.05
CRI-Education	111.6 ± 16.9	113.5 ± 15.1	110.7 ± 18.4	*p* > 0.05
CRI-Working Activity	98.9 ± 20.4	102.3 ± 21.9	96.6 ± 18.9	***p* < 0.05**
CRI-Leisure Time	106.22 ± 22.1	106.3 ± 23.1	106.2 ± 21.4	*p* > 0.05
CRI-Total Score	107.4 ± 30.1	109.2 ± 24.1	106.2 ± 33.7	*p* > 0.05

Legend: CRI = Cognitive Reserve Index; OT = Odor Threshold; OD = Odor Discrimination; OI = Odor Identification; SD = standard deviation; TDI score = Threshold, Discrimination, and Identification score. Bold indicates statistical differences of *p* < 0.05 between men and women.

**Table 2 biology-12-00586-t002:** Bivariate correlations between olfactory function (OT, OD, OI, and TDI) and each specific sub-score of the CRI.

		CRI-Education	CRI-Working Activity	CRI-Leisure Time	CRI-Total Score
OT	*r*	−0.150	0.066	0.058	0.782
	*p*	**0.014**	0.282	0.340	0.269
OD	*r*	0.077	0.408	0.364	0.197
	*p*	0.206	**0.001**	**0.001**	**0.001**
OI	*r*	0.176	0.406	0.389	0.337
	*p*	**0.004**	**0.001**	**0.001**	**0.001**
TDI	*r*	0.063	0.308	0.304	0.298
	*p*	0.301	**0.001**	**0.001**	**0.001**

Legend: CRI = Cognitive Reserve Index; OT = Odor Threshold; OD = Odor Discrimination; OI = Odor Identification; TDI score = Threshold, Discrimination, and Identification score. Bold indicates statistical differences of *p* < 0.05.

**Table 3 biology-12-00586-t003:** Multivariate linear regression analysis models using OT, OD, and OI as dependent variables and each specific sub-score of CRI as independent variables.

	Unstandardized Coefficients		Standardized Coefficients		
	B	Std Error	β	t	*p*
Model (1) OT as a dependent variable					
CRI-Education	−0.045	0.015	−0.211	−3.076	**0.002**
CRI-Working Activity	0.014	0.014	0.079	0.990	0.323
CRI-Leisure Time	0.008	0.013	0.049	0.622	0.535
Model (2) OD as a dependent variable					
CRI-Education	−0.010	0.012	−0.052	0.835	0.404
CRI-Working Activity	0.503	0.011	0.349	4.784	**0.001**
CRI-Leisure Time	0.033	0.010	0.235	3.279	**0.001**
Model (3) OI as a dependent variable					
CRI-Education	0.001	0.011	−0.002	−0.034	0.973
CRI-Working Activity	0.033	0.010	0.236	3.235	**0.001**
CRI-Leisure Time	0.026	0.009	0.201	2.816	**0.005**

Legend: CRI = Cognitive Reserve Index; OT = Odor Threshold; OD = Odor Discrimination; OI = Odor Identification; Std error = standard error. Bold indicates statistical differences of *p* < 0.05.

**Table 4 biology-12-00586-t004:** Bivariate correlations between olfactory function (OT, OD, OI, and TDI) and each specific sub-score of CRI in women (*n* = 158).

		CRI-Education	CRI-Working Activity	CRI-Leisure Time	CRI-Total Score
OT	*r*	0.001	0.111	0.278	0.130
	*p*	0.990	0.166	**0.001**	0.103
OD	*r*	0.135	0.313	0.398	0.325
	*p*	0.092	**0.001**	**0.001**	**0.001**
OI	*r*	0.184	0.426	0.546	0.337
	*p*	**0.020**	**0.001**	**0.001**	**0.001**
TDI	*r*	0.333	0.355	0.526	0.298
	*p*	**0.001**	**0.001**	**0.001**	**0.001**

Legend: CRI = Cognitive Reserve Index; OT = Odor Threshold; OD = Odor Discrimination; OI = Odor Identification; TDI score = Threshold, Discrimination, and Identification score. Bold indicates statistical differences of *p* < 0.05.

**Table 5 biology-12-00586-t005:** Multivariate linear regression analysis models using OT, OD, OI, and each specific sub-score of CRI as independent variables in women (*n* = 158).

	Unstandardized Coefficients		Standardized Coefficients		
	B	Std Error	β	t	*p*
Model (1) OT as a dependent variable					
CRI-Education	−0.014	0.017	−0.073	−0.842	0.401
CRI-Working Activity	−0.005	0.016	−0.029	−0.294	0.769
CRI-Leisure Time	0.063	0.020	0.319	3.175	**0.002**
Model (2) OD as a dependent variable					
CRI-Education	−0.004	0.011	−0.028	−0.344	0.731
CRI-Working Activity	0.014	0.011	0.114	1.235	0.219
CRI-Leisure Time	0.040	0.014	0.280	2.943	**0.004**
Model (3) OI as a dependent variable					
CRI-Education	0.001	0.011	0.007	0.100	0.920
CRI-Working Activity	0.025	0.011	0.201	2.398	**0.018**
CRI-Leisure Time	0.071	0.013	0.463	5.372	**0.001**

Legend: CRI = Cognitive Reserve Index; OT = Odor Threshold; OD = Odor Discrimination; OI = Odor Identification. Bold indicates statistical differences of *p* < 0.05.

**Table 6 biology-12-00586-t006:** Bivariate correlations between olfactory function (OT, OD, OI, and TDI) and each specific sub-score of CRI in men (*n* = 111).

		CRI-Education	CRI-Working Activity	CRI-Leisure Time	CRI-Total Score
OT	*r*	−0.346	−0.005	−0.105	−0.070
	*p*	**0.001**	0.959	0.111	0.464
OD	*r*	0.164	0.379	0.333	0.043
	*p*	0.085	**0.001**	**0.001**	0.652
OI	*r*	0.194	0.342	0.334	0.387
	*p*	**0.042**	**0.001**	**0.001**	**0.001**
TDI	*r*	−0.016	0.290	0.205	0.244
	*p*	0.869	**0.002**	**0.031**	**0.010**

Legend: CRI = Cognitive Reserve Index; OT = Odor Threshold; OD = Odor Discrimination; OI = Odor Identification; TDI score = Threshold, Discrimination, and Identification score. Bold indicates statistical differences of *p* < 0.05.

**Table 7 biology-12-00586-t007:** Multivariate linear regression analysis models of OT, OD, and OI as dependent variables and each specific sub-score of CRI as independent variables in men (*n* = 111).

	Unstandardized Coefficients		Standardized Coefficients		
	B	Std Error	β	t	*p*
Model (1) OT as a dependent variable					
CRI-Education	−0.095	0.028	−0.404	−3.399	**0.001**
CRI-Working Activity	0.018	0.021	0.119	0.865	0.389
CRI-Leisure Time	−0.027	0.023	−0.166	−1.150	0.253
Model (2) OD as a dependent variable					
CRI-Education	−0.011	0.023	−0.057	−0.488	0.626
CRI-Working Activity	0.026	0.017	0.206	1.516	0.132
CRI-Leisure Time	0.012	0.019	0.088	0.615	0.540
Model (3) OI as a dependent variable					
CRI-Education	−0.009	0.023	−0.047	−0.398	0.691
CRI-Working Activity	0.014	0.017	0.107	0.781	0.437
CRI-Leisure Time	0.009	0.019	0.069	0.480	0.632

Legend: CRI = Cognitive Reserve Index; OT = Odor Threshold; OD = Odor Discrimination; OI = Odor Identification; Std Error = standard error. Bold indicates statistical differences of *p* < 0.05.

## Data Availability

Not applicable.

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
