# Peer review of "Gender-Related Differences in the Correlation between Odor Threshold, Discrimination, Identification, and Cognitive Reserve Index in Healthy Subjects"

_biology, 2023, doi:10.3390/biology12040586_

Round 1

Reviewer 1 Report

The study examines an important, yet still understudied link between olfactory performance and cognitive functions and seeks to explore potential gender-related differences. Authors identify more associations in women than in men. They conclude that olfactory impairments may serve as a biomarker for cognitive impairment. 

The manuscript is well-written and it does make an important contribution to the current knowledge. Before acceptance, I would suggest making the following changes: 

1) page 2, line 49 - "sensorial" is unclear, perhaps you can rephrase?

2) page 2, lines 74-75 - BMI is a combined index for height and weight, no need to inform that you collected height, weight and BMI

3) page 2, line 82 - a "." is missing at the end of the section

4) page 3, line 95 - please use the latest Sniffin Sticks normative data

5) page 4, line 145 - PD patients appear, but they are not mentioned in the participants section (2.1)?

6) page 11, lines 301 - 307 - I do not understand this argument, perhaps you can rephrase it

7) page 12, lines 361 - 362 - you could give some recommendations what evidence is needed to fully address the causal link between olfactory loss and mild cognitive impairment (MCI). Importantly, the discussion lacks a clear conclusion on how the observed gender differences in these links can be used in managing/predicting MCI. 

General comments: 

I wonder if there is a need for analyzing TDI score, if you provide analyses for OT, OD and OI separately, as the TDI is a sum of these three subscales. It is redundant and removing the models examining TDI would make the manuscript shorted and clearer. The same applied for the total CRI score.

In the materials and methods section you may want to describe CRI in more detail - for instance how it is scored, what are the possible scores?

Authors found that men and women differ in CRI-Working Activity scores. Is it possible that this difference is produced by different age when people retire?

The manuscript could be written and proofed by a native speaker, as it contains some odd phrases (e.g. "under this scenario" page 2, line 59)

Author Response

The study examines an important, yet still understudied link between olfactory performance and cognitive functions and seeks to explore potential gender-related differences. Authors identify more associations in women than in men. They conclude that olfactory impairments may serve as a biomarker for cognitive impairment.

The manuscript is well-written and it does make an important contribution to the current knowledge. Before acceptance, I would suggest making the following changes:

1) page 2, line 49 - "sensorial" is unclear, perhaps you can rephrase?

Answer 1) Authors thank the Reviewer for this suggestion and revised the sentence as indicated: “The OT is usually more associated to the individual differences of nasal anatomy [11], and it is related to specific cognitive domain such as the language function [6].”

2) page 2, lines 74-75 - BMI is a combined index for height and weight, no need to inform that you collected height, weight and BMI

Answer 2) Authors revised the text according to the Reviewer’ suggestion.

3) page 2, line 82 - a "." is missing at the end of the section

Answer 3) Authors included this correction on the text according to the Reviewer’ suggestion.

4) page 3, line 95 - please use the latest Sniffin Sticks normative data

Answer 4) Authors thank the Reviewer for the observation and revised the text into the following sentence: “Then, we calculated the total score of olfactory function (Threshold, Discrimination, Identification, TDI), functional anosmia, hyposmia, normosmia, supersmellers were indicated by ≤ 16, between 16.25 and 30.5, a score between 30.75 and 41.25, a score > 41.5, respectively [20].” This correction is in red color and underlined in yellow on the text.

5) page 4, line 145 - PD patients appear, but they are not mentioned in the participants section (2.1)?

Answer 5) Authors apologised for this mistake and revise the text In this study we did not involve any patient with Parkinson’s disease.

6) page 11, lines 301 - 307 - I do not understand this argument, perhaps you can rephrase it

Answer 6) These sentences of the Discussion section is now revised on the text. The corrections are in red color and underlined in yellow on the text.

7) page 12, lines 361 - 362 - you could give some recommendations what evidence is needed to fully address the causal link between olfactory loss and mild cognitive impairment (MCI). Importantly, the discussion lacks a clear conclusion on how the observed gender differences in these links can be used in managing/predicting MCI.

Answer 7) In line to Reviewer suggestion we have revised and implemented the Conclusion section with the following sentence: “The gender differences observed in our study could play a key role in order to predict the risk of MCI and to develop a precision medicine approach.”

General comments:

8) I wonder if there is a need for analyzing TDI score, if you provide analyses for OT, OD and OI separately, as the TDI is a sum of these three subscales. It is redundant and removing the models examining TDI would make the manuscript shorted and clearer. The same applied for the total CRI score.

Answer 8) Authors in line to the Reviewer’s suggestion decided to delete TDI score and CRI Total Score in the description of multivariate linear regression analyses. Consequently, the Tables 3,4, and 7 are now changed in the text. In addition, the Figures 1 and 2 are now changed in the Manuscript.

9) In the materials and methods section you may want to describe CRI in more detail - for instance how it is scored, what are the possible scores?

Answer 9) Authors implemented the description of CRI test in the Materials and Methods section.

The possible scores for CRI-Education, CRI-Working Activity, and CRI-Leisure Time are different in relation to age and gender as previously indicated by Nucci and Colleagues (2012). The study of Nucci and Colleagues (2012) indicated mean scores for men and women in different age ranges as following indicated:

10) Authors found that men and women differ in CRI-Working Activity scores. Is it possible that this difference is produced by different age when people retire?

Answer 10) The difference in CRI-Working Activity scores between men and women may be due to different age when people retire. Authors in line with Reviewer’s suggestion implemented this section in the Discussion.

11) The manuscript could be written and proofed by a native speaker, as it contains some odd phrases (e.g. "under this scenario" page 2, line 59).

Answer 11) The Manuscript has been revised by a native speaker and all corrections are in red color and underlined in yellow on the text.

Reviewer 2 Report

Thank you for the possibility of reviewing this paper. The topic addressed by the authors is relevant, and I am pleased to add something to the manuscript. I hope the authors will find my suggestions helpful.

l. 11 Many studies indicated significant associations between olfactory function and cognitive abilities in healthy controls – could you please be more specific? What kind of associations?

l. 20 – again, could you please be more specific?

l. 26-27 “considering all the subjects…” – please, rephrase the passive voice in this sentence

l. 30 was instead of were

l. 40 please, add references to enforce your statement

l 45 did you mean ‘has increased significantly’?

l 70 please, describe mean age and standard deviation for each gender separately

l 93 please, rephrase the passive voice

l 97 could you please give an exemplary item from each scale (education, etc.)?

l 119 please, delete the coma before the bracket

l 136 please, present the definition of TDI when you first use it

Table 3. 4 and 7; I am not sure if I follow the logic of regression analyses. Did you insert all CRI scale’s components (education, etc.) as independent variables? Did you check if they are correlated with each other? In case they are, and I would assume so, it can affect your results

L 186 rephrase the passive voice

Author Response

Thank you for the possibility of reviewing this paper. The topic addressed by the authors is relevant, and I am pleased to add something to the manuscript. I hope the authors will find my suggestions helpful.

  1. Line 11 Many studies indicated significant associations between olfactory function and cognitive abilities in healthy controls – could you please be more specific? What kind of associations?

Answer 1) Authors implemented this section in the Introduction according to the Reviewer suggestion.

  1. Line 20 – again, could you please be more specific?

Answer 2) Authors included this correction on the text in the Abstract section according to the Reviewer’ suggestion. The corrections are in red color and underlined in yellow on the text.

  1. Line 26-27 “considering all the subjects…” – please, rephrase the passive voice in this sentence

Answer 3) This sentence is now changed into the following: “In all subjects’ significant associations between the odor threshold versus CRI-Education, between the odor discrimination and identification versus CRI-Working activity and CRI-Leisure Time, were found.”

  1. Line 30 was instead of were

Answer 4) Authors thank the Reviewer for this suggestion and revised the text.

  1. Line 40 please, add references to enforce your statement

Answer 5)This section has been revised on the text in line to the Reviewer’s suggestion.

  1. Line 45 did you mean ‘has increased significantly’?

Answer 6) This sentence is now changed into the following: “Recently, the importance of olfactory assessment has much increased particularly in cases of cognitive decline and COVID-19.”

  1. Line 70 please, describe mean age and standard deviation for each gender separately

Answer 7) The description of mean age and standard deviation for each gender has now included in the text in line with this suggestion.

  1. Line 93 please, rephrase the passive voice

Answer 8) This sentence is now changed into the following: “Then, we calculated the total score of olfactory function (Threshold, Discrimination, Identification, TDI), functional anosmia, hyposmia, normosmia, supersmellers were indicated by a score ≤ 16, between 16.25 and 30.5, between 30.75 and 41.25, > 41.5, respectively [20].” This correction is in red color and underlined in yellow on the text.

9) Line 97 could you please give an exemplary item from each scale (education, etc.)?

Answer 9) Authors implemented the description of CRI test in the Materials and Methods section.

The possible scores for CRI-Education, CRI-Working Activity, and CRI-Leisure Time are different in relation to age and gender as previously indicated by Nucci and Colleagues (2012). The study of Nucci and Colleagues (2012) indicated mean scores for men and women in different age ranges as following indicated:

 10) Line 119 please, delete the coma before the bracket

Answer 10) Authors thank the Reviewer for this suggestion and included this correction on the text.

11) Line 136 please, present the definition of TDI when you first use it.

Answer 11) Authors thank the Reviewer for this suggestion and included this correction on the text.

12) Table 3. 4 and 7; I am not sure if I follow the logic of regression analyses. Did you insert all CRI scale’s components (education, etc.) as independent variables? Did you check if they are correlated with each other? In case they are, and I would assume so, it can affect your results.

Answer 12) Authors in line to the Reviewer 1 and 2 decided to delete TDI score and CRI Total Score in the models of multivariate linear regression analyses since they were the sum of each specific sub-score. Consequently, the Tables 3,4, and 7 are now changed in the text. In addition, the Figures 1 and 2 are now changed in the Manuscript. Multivariate linear regression analyses were performed using OT, OD, and OI as dependent variables and each specific sub-score of CRI as independent variables for total subjects, women and men.

13) Line 186 rephrase the passive voice

Answer 13) Authors thank the Reviewer for this suggestion and included this correction on the text.

Reviewer 3 Report

Thank you for the opportunity to review this manuscript. While I think the topic may be important and make a contribution to the literature, I found it very difficult to assess as the paper is currently written.

The major issue that I see is that the motivation/Introduction is underdeveloped. The Introduction is too short and the rationale for all of the many, many analyses is unclear. Some of the motivation is made clearer in the Discussion, but that seemed too late. Fundamentally, and this may be my ignorance, the motivation to use the CRI is unclear. It is not measuring cognitive ability per se, but, to the extend that the paper suggests a motivation to study a link between the CRI and olfaction, it seems to be about the link between cognitive ability and olfactory function.

Given the lack of clear motivation and any sort of hypotheses, it is unclear to me how one should think about any correlations between the CRI or the sub-scales of the CRI and olfactory function. Why should CRI-Leisure have anything to do with one measure of olfaction in one gender, for example? This is not made any clearer in the Discussion and, in fact, some of the results seem to surprise the authors. What were their expectations? Predictions? Hypotheses? And based on what theory would they make these predictions? I'm left no further ahead in my understanding of the relationship between cognitive and olfactory processing/function. I am definitely unclear about how the authors arrive at this conclusion: "... our data suggest(ed) a potential role of biomarker for olfactory function in the early diagnosis of mild cognitive impairment." (p. 12) How is this the case? I found this particularly confusing given that all participants were healthy. (I did note some mention of PD patients in the Results, which seemed to come out of the blue - p. 4.)

Overall, this paper would really need to be re-written to explain the motivation and help the reader understand all of the many correlational analyses.

This may be a typo, but the authors indicate that they began collecting data in 2017 but that the IRB protocol was approved in 2018. This needs to be corrected or explained.

In places the English is a little bit hard to follow.

Author Response

Thank you for the opportunity to review this manuscript. While I think the topic may be important and make a contribution to the literature, I found it very difficult to assess as the paper is currently written.

  • The major issue that I see is that the motivation/Introduction is underdeveloped. The Introduction is too short and the rationale for all of the many, many analyses is unclear. Some of the motivation is made clearer in the Discussion, but that seemed too late. Fundamentally, and this may be my ignorance, the motivation to use the CRI is unclear. It is not measuring cognitive ability per se, but, to the extend that the paper suggests a motivation to study a link between the CRI and olfaction, it seems to be about the link between cognitive ability and olfactory function. Given the lack of clear motivation and any sort of hypotheses, it is unclear to me how one should think about any correlations between the CRI or the sub-scales of the CRI and olfactory function.

Answer 1) Authors revised the Introduction and Discussion sections in order to better explain the rationale of this study. In the Introduction we included the following sections:

“In literature have been observed recent advances on cognitive abilities studies in relation to well-being, social interactions, and sport activities; in particular, some re-search focused on the role of chemosensory perception [1,2]. In fact, olfactory function could act on cognitive abilities as related to sematic memory, executive function, and verbal fluency [3-5]. Moreover, a low language ability has been associated to low olfactory function [3-5]. Olfactory function plays important role in the regulation of emotional function, social interactions, and eating behavior. The decreased olfactory function may show negative implications for human well-being and social relationships. Olfactory function and cognitive performance showed an overlap in neural substrates and were associated to orbitofrontal cortex, hippocampus, and entorhinal cortex [3-5]. This relationship suggested the need to evaluate olfactory function as a potential early biomarker in patients’ cognitive impairment [6-9]. Recently, the importance of olfactory assessment has much increased particularly in cases of mild cognitive decline and in the diagnosis of Covid-19.

The objective evaluation of the quantitative olfactory dysfunction comprises three different components: odor threshold (OT), odor discrimination (OD), and odor identification (OI) [10]. The OT is usually more associated to the individual differences of nasal anatomy [11], and it is related to specific cognitive domain such as the language function [6]. Indeed, OD and OI are considered in relation to subcortical part of the olfactory system and are related to the ability to differentiate and to identify different odors, respectively [12]. Usually, lower scores in cognitive tests associated with a deficit in olfactory function may suggest a mild cognitive impairment [13].

The brain cognitive reserve is the potential ability of the brain to cope neuronal damage.

Cognitive reserve plays an important role in order to recover brain damage affected by aging or neurodegenerative diseases through the recruitment of pre-existing brain networks. In general, setting aside neurocognitive resources during the life, may preserve from mild cognitive impairment and pathologies as Alzheimer disease [14,15]. Since around 65% of subjects affected by Alzheimer disease are women [16], understanding gender-related differences, in the relationship between olfactory function and the specific cognitive domain of the CRI questionnaire, appears important in order to obtain most incisive interventions for an early diagnosis of mild cognitive impairment and Alzheimer disease. In literature, few studies evaluated gender-differences in CRI [17,18], but the gender-related association between olfactory function and each specific cognitive domain of the CRI questionnaire have not been evaluated so far.

The aim of this study was first to evaluate to gender-related differences in each specific cognitive domain of the CRI questionnaire such as education, working activity, and leisure time, in healthy subjects. Then, we evaluated gender-related differences in the relationship between olfactory function and in each specific cognitive domain of the CRI.”

All corrections in the Introduction and Discussion are in red color and underlined in yellow on the text.

  • Why should CRI-Leisure have anything to do with one measure of olfaction in one gender, for example? This is not made any clearer in the Discussion and, in fact, some of the results seem to surprise the authors. What were their expectations? Predictions? Hypotheses? And based on what theory would they make these predictions? I'm left no further ahead in my understanding of the relationship between cognitive and olfactory processing/function. I am definitely unclear about how the authors arrive at this conclusion: "... our data suggest(ed) a potential role of biomarker for olfactory function in the early diagnosis of mild cognitive impairment." (p. 12) How is this the case? I found this particularly confusing given that all participants were healthy. (I did note some mention of PD patients in the Results, which seemed to come out of the blue - p. 4.)

Answer 2) The olfactory deficit is considered an important health issue (Murphy et al., 2002), which could play an important role in humans social interactions and well-being.

Our study suggested the presence of complex relationships between olfactory function and cognitive abilities as previously reported by other studies:

  • Oleszkiewicz, A.; Kunkel, F.; Larsson, M.; Hummel, T. Consequences of undetected olfactory loss for human chemosensory communication and well-being. Phil Trans R Soc B, 2020, 375, 20190265. http://dx.doi.org/10.1098/rstb.2019.0265
  • Kostka, J.K.; Bitzenhofer, S.H. How the sense of smell influences cognition throughout life. Neuroforum 2022, 28(3), 177-185.
  • Eibenstein, A.; Fioretti, A.B.; Simaskou, M.N.; Sucapane, P.; Mearelli, S.; Mina, C.; et al. Olfactory screening test in mild cognitive impairment. Neurol Sci 2005, 26(3), 156–60.
  • Ottaviano, G.; Frasson, G.; Nardello, E.; Martini, A. Olfaction deterioration in cognitive disorders in the elderly. Aging Clin Exp Res 2016, 28(1), 37–45.

The strong relationship between olfaction and cognitive functions may be explained the tight anatomical and functional coupling between these two systems. The lateral entorhinal cortex (LEC) plays a key role in the interaction between the olfactory and the cognitive systems as indicated by Kostka and Bitzenhofer (Neuroforum 2022; 28(3): 177–185).

A Figure probably may better explain brain connections between olfaction and cognitive functions (as previously reported by Kostka and Bitzenhofer, 2022).

In old age and in elderly subjects is very common to find an olfactory deficit associated with cognitive impairment (Attems et al., 2015; Murman,2015; Uchida et al., 2020; Yahiaoui-Doktor et al., 2019).

In this context, the brain cognitive reserve is closely related to cortical plasticity and is considered the potential capability of the brain to cope neuronal damage in relation to individual differences such as brain size and synapse count as indicated by Nucci and Colleagues (2012). The CRI questionnaire consists of 20 items grouped in three main sources: education, working activity, and leisure time. Each of these items of a subject’s lifetime is calculated as a sub-score. The CRI-Education evaluated years of education and the possible training courses lasting at six months and the total score was the sum of these factors. The CRI-Working Activity evaluated the number of the years on each profession during the lifespan and the total score was the sum of the working activity years multi-plied by cognitive level of job from 1 to 5 years. The CRI-Leisure Time evaluates the frequency and the number of years spent in intellectual activities. The total score was computed as the total number of the years involved in these activities in which frequency is often/always.

As indicated in the Discussion of our Manuscript, “the relationship between OI and OD versus CRI-Leisure Time is not clear to understand. In the multivariate linear regression analyses we found that there was a positive significant association between OI and OD versus CRI-Leisure Time only in women, but not in men. This data supports the hypothesis that CRI-Leisure Time is closely correlated to the OI and OD performance only in women. Generally, women showed better olfactory performance compared to men [10,37] and, also, had more social connections. Probably, women perform differently in social relationships and could be possible an association between leisure time, social network, and health measures. In fact, Codina and Pestana showed that men had more leisure time, but women had a high positive leisure experience than men [38]; women enjoyed them-selves more with less leisure time and were more positive to time orientations. Moreover, Larsson and colleagues observed that women identified more odors than men, but sex differences disappeared in the oldest age [35]. Although the potential cause of the difference between men and women remains unclear, probably the higher identification in women may be due to sex differences in verbal abilities, prior experience, and the modulating role of sex hormones on olfactory behavior [39].”

Women showed better olfactory perception compared to men due to gonadal hormones, menstrual cycle-related fluctuations, and neuroendocrine influences on brain regions involved in olfactory function (Doty & Cameron, 2009; Sorokowski et al., 2019). However, social factors may also contribute as women generally experience greater olfactory pleasantness, odor familiarity, and greater exposure to odors in their social environment.

Previous studies indicated that subjects with larger social networks and social relationships (such as friends and co-worker) showed better cognitive health (Cohn-Schwartz et al., 2021; Goldman, 2023).

Finally, regards page 4, line 145 - PD patients, Authors apologised for this mistake and revise the text. In this study we did not involve any patient with Parkinson’s disease.

  • Overall, this paper would really need to be re-written to explain the motivation and help the reader understand all of the many correlational analyses.

Answer 3) Authors, in line with the Reviewers suggestion revised the Introduction, Material and Methods, Discussion, and Conclusion of this Manuscript. All corrections in the Introduction and Discussion are in red color and underlined in yellow on the text.

  • This may be a typo, but the authors indicate that they began collecting data in 2017 but that the IRB protocol was approved in 2018. This needs to be corrected or explained.

Answer 4) Authors apologised for this mistake and revise the text. The sentence is now revised into the following: “Data were collected from May 2018 to January 2020.”

5) In places the English is a little bit hard to follow.

Answer 5) The Manuscript has been revised by a native speaker and all corrections are in red color and underlined in yellow on the text.

Round 2

Reviewer 2 Report

Dear Authors,

I am glad you found my comments helpful. I have no more remarks,

Best wishes

Author Response

Authors thank the Reviewer for his helpful suggestions

Reviewer 3 Report

None.

Author Response

Authors thank the Reviewer 3.